# CHASM: Unveiling Covert Advertisements on Chinese Social Media

**Jingyi Zheng**[1]* **Tianyi Hu**[2]*
**Yule Liu**[1]   **Zhen Sun**[1]   **Zongmin Zhang**[1]   **Zifan Peng**[1]   **Wenhan Dong**[1]†   **Xinlei He**[1]
[1]Hong Kong University of Science and Technology (Guangzhou)   [2] Aarhus Univeristy

## Abstract

Current benchmarks for evaluating large language models (LLMs) in social media moderation completely overlook a serious threat: covert advertisements, which disguise themselves as regular posts to deceive and mislead consumers into making purchases, leading to significant ethical and legal concerns. In this paper, we present the **CHASM**, a first-of-its-kind dataset designed to evaluate the capability of Multimodal Large Language Models (MLLMs) in detecting covert advertisements on social media. CHASM[3] is a high-quality, anonymized, manually curated dataset consisting of 4,992 instances, based on real-world scenarios from the Chinese social media platform Rednote. The dataset was collected and annotated under strict privacy protection and quality control protocols. It includes many product experience sharing posts that closely resemble covert advertisements, making the dataset particularly challenging. The results show that under both zero-shot and in-context learning settings, none of the current MLLMs are sufficiently reliable for detecting covert advertisements. Our further experiments revealed that fine-tuning open-source MLLMs on our dataset yielded noticeable performance gains. However, significant challenges persist, such as detecting subtle cues in comments and differences in visual and textual structures. We provide in-depth error analysis and outline future research directions. We hope our study can serve as a call for the research community and platform moderators to develop more precise defenses against this emerging threat.

## 1   Introduction

Social media platforms offer users spaces to create and share content [1], and social media advertising has become one of the most successful forms of internet marketing, influencing billions of consumers worldwide [2]. This thriving economy benefits not only social media platforms but also content creators and advertisers [3]. However, people are tired of the many advertisements on social media and are likely to skip them [4]. Covert advertisements have emerged and spread widely to capture user attention, raising significant public concern. As shown in Figure 1, unlike traditional advertisements, covert advertisements are deliberately designed to resemble regular content [5], such as product experience sharing, to subtly persuade unsuspecting viewers to purchase the featured products.

Despite its benefits for consumer engagement, its inherently deceptive nature has sparked widespread public criticism [6], such as consumer fraud [7], damage to the platform's credibility [8], and harmful

---

*Equal contribution

†Corresponding author

[3]The Dataset is available at `https://huggingface.co/datasets/Jingyi77/CHASM-Covert_Advertisement_on_RedNote`, and the Code is available at `https://github.com/Jingyi62/CHASM`

39th Conference on Neural Information Processing Systems (NeurIPS 2025) Track on Datasets and Benchmarks.

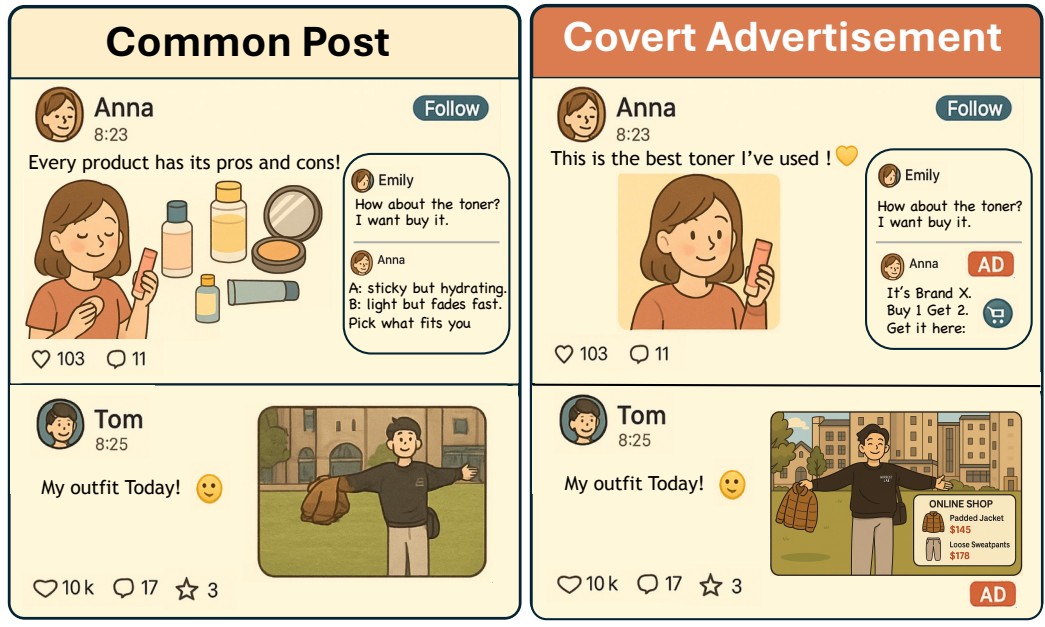

Figure 1: Typical examples of covert advertisement. Although it appears very similar to the common lifestyle-sharing posts on the left, the covert advertisements on the right promote products through implicit signals, such as hidden cues in the image or the comment section. The concealment and diverse variations of covert advertisements make detecting them particularly challenging.

effects on users' consumption habits [9]. This has led covert advertisements to raise both ethical and legal concerns: on one hand, they gain an unfair advantage in commercial competition through deception; on the other hand, they violate laws in many countries, such as China and the United States [10, 11], that require advertisements to be clearly identifiable to consumers.

Given the large scale of new content generated on social media platforms, LLMs and MLLMs have been widely adopted as a scalable and efficient tool for content moderation on social media [12, 13], providing users with a better community environment while significantly reducing the costs associated with manual review. However, existing research mainly focuses on regulating other harmful content on social media, such as fake news [14, 15], cyberbullying [16], toxic content [17], and hate speech [18, 19]. Covert advertisements, which can likewise carry substantial negative impacts and clearly violate laws, remain largely unexplored. To the best of our knowledge, no existing MLLMs have been trained to detect covert advertisements, nor are there publicly available datasets or task guidelines to facilitate the training and evaluation of such models.

Different from the detection of other harmful content on social media, regulating covert advertisements presents several unique challenges. First, covert advertisements may appear in either text or images, making the task inherently multimodal. Second, advertisers deliberately conceal their intent, resulting in a high degree of stealth. Third, social media naturally contains many real user posts sharing shopping experiences, which are easily mistaken for advertisements, further increasing the difficulty of distinguishing covert advertisements.

To address these issues, we proposed CHASM: Covert Hype Advertisement in Social Media. CHASM is a first-of-its-kind, high-quality, strictly privacy-preserving, and manually curated challenging dataset grounded in real-world scenarios. The data is sourced from the RedNote platform [4] and consists of real-world posts, including post content, images, and associated comments. Our dataset deliberately includes many real, non-advertisement posts that closely resemble covert advertisements, such as user sharing of shopping experiences or product usage, to reduce the risk of misclassifying normal product-sharing content, which makes detecting covert advertisements particularly challenging. Data collection strictly adheres to the platform's user agreement, including policies on user

---

[4]RedNote (https://www.xiaohongshu.com) is one of the most popular social platforms in China, with over 120 million daily active users

privacy protection and copyright regulations. Additional anonymization measures are taken to protect user privacy. We adopt a dynamic quality control annotation framework, incorporating pre-designed gold-standard questions and a three-annotator majority voting mechanism for difficult cases, resulting in high-quality annotations.

Using CHASM, we conducted systematic evaluations of various LLMs, including the state-of-the-art MLLMs such as GPT-4o [20] and DeepSeek-V3 [21], smaller-scale open-source LLMs such as LLaVA [22] and Qwen2.5-7B [23], as well as the latest reasoning MLLMs, such as Gemini2.5 Pro [24]. Our experimental results show that most tested models struggle with the task under both zero-shot and in-context learning settings. GPT-4o achieved the best baseline performance of only 59.7% F1-Score, even MLLMs with strong reasoning capabilities are not sufficient to yield a significant advantage on our task. Further exploration shows that fine-tuning open-source MLLMs on our dataset leads to substantial performance improvements. Notably, Qwen2.5-7B achieved an F1-Score of 75.6%, significantly surpassing the zero-shot state-of-the-art, empirically showing the effectiveness of our dataset. By analyzing the types of errors made across all different settings, We find that fine-tuning notably improves the model's grounding in factual evidence. However, the fine-tuned models still struggle with recognizing visual and textual structural features, as well as detecting subtly embedded advertisements. These results can provide insights into future improvements in the covert advertisement detection capabilities of MLLMs.

Our contributions can be summarized as follows:

- We propose a new task of detecting covert advertisements. We analyze key challenges and provide detailed assessment guidelines with clear criteria and examples.
- We manually curated CHASM, a novel dataset for evaluating the capabilities of MLLMs in detecting covert advertisements, based on challenging real-world cases from RedNote.
- We conducted comprehensive evaluations on CHASM using various open- and closed-source MLLMs, finding that none of the current MLLMs are sufficiently reliable for detecting covert advertisements under either zero-shot or in-context learning settings. Fine-tuning open-source MLLMs on our dataset leads to significant improvements in performance.
- Our error analysis reveals the limitations of even fine-tuned MLLMs, including their difficulty in recognizing visual and textual structural features as well as detecting subtly embedded advertisements. We also provide concrete directions for platform moderators to improve the detection of covert advertisements.

## 2 The Task of Covert Advertisement Detection

In this section, we propose a novel task: covert advertisement detection on social media. We define key characteristics that covert advertisements should possess in Section 2.1, highlight the main challenges in detecting them, and provide guidelines to assist in judgment in Section 2.2.

### 2.1 Task Definition

Drawing inspiration from previous marketing research [25–28], our formal definition of the covert advertisement is as follows:

**Definition 1** *Covert advertisement is promotional content made to look like common content with the primary aim of subtly influencing the audience's consumption decisions without explicitly disclosing its advertising nature.*

Covert advertisements must meet two key criteria: First, the author must have a clear intent to promote a product or paid service for direct financial gain from the associated brand. Here, *profit* is narrowly defined as monetary compensation, excluding indirect benefits like persuasion or follower growth. Second, the author must deliberately disguise the post to resemble regular content. Posts clearly labeled as ads by the platform or user are not considered covert advertisements.

We acknowledge that the criteria for covert advertisements are subjective. For example, some regular product experience posts may also contain "praising" language, and the distinction between such praise and promotional exaggeration can vary from person to person. To mitigate annotation inconsistencies caused by this subjectivity, we further specified the evidence-driven guideline in Section 2.2 and Appendix A as a reference, and employed a majority voting mechanism in Section 3.1 to resolve disputed cases.

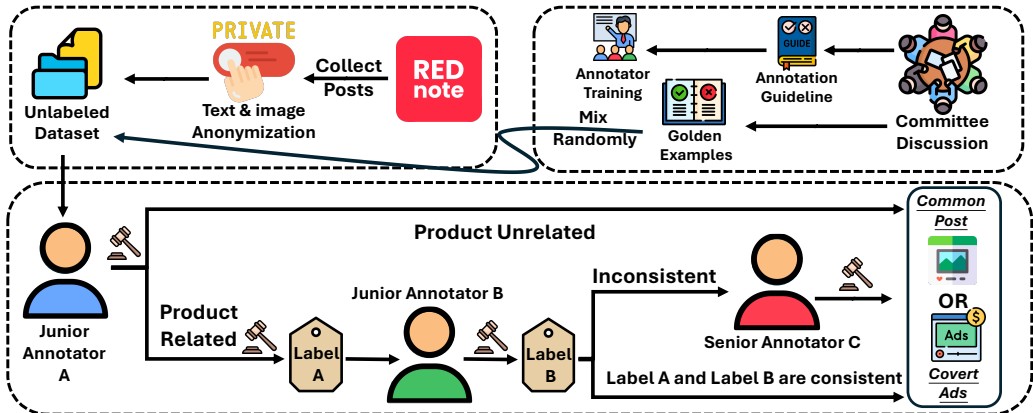

Figure 2: The construction of CHASM follows a three-stage process: (1) Data collection and anonymization, (2) Committee-driven curation of guidelines and gold questions, (3) Difficulty-aware dynamic annotation workflow. These stages ensure that the dataset maintains strict privacy protection, includes challenging product-sharing examples, and achieves high-quality annotations.

## 2.2 Main Challenges and Guidelines

Social media is filled with lifestyle content, where product-related posts often appear in contexts like travel, daily routines, and food. However, since much of this content reflects personal experience, it's unreasonable to assume all such posts are advertisements. The main challenge in covert advertisement detection is distinguishing *genuine product sharing* from content with hidden promotional intent (*covert advertisements*).

Given the deceptive nature of covert advertisements and the subjective line between them and genuine product sharing, annotations can be ambiguous. To reduce this ambiguity and improve consistency in both human and model judgments, we propose a set of systematic, evidence-based guidelines for detecting covert advertisements:

**Clear Promotional Evidence**: Covert advertisements often include clear signs of promotion, such as providing direct purchase links or instructions on buying the product. To make the advertisement more covert, promotional links are sometimes embedded in images or comments, or users are redirected to private chat groups for sales. In contrast, non-advertising content is primarily focused on sharing personal experiences, and thus may only casually mention the product or store name, and the content often lacks sufficient information for users to complete a purchase.

**Language Style of Posts**: Covert advertisements often adopt clickbait-style headlines and sales talk. The writing typically carries a strong promotional tone, using exaggerated language to emphasize the product's benefits, which deviates from the natural style of everyday communication. In contrast, non-advertising content usually maintains a more casual tone and focuses on sharing personal experiences rather than promoting a product. It may also include mentions of the product's shortcomings.

**Text and Image Structure of Posts**: Covert advertisements typically focus their text and images on a single specific product or closely related products from the same brand. In contrast, non-promotional lifestyle sharing posts often feature multiple different brands within the same category, some of which may even be competitors, or the author does not explicitly advocate any particular brand.

A more detailed guideline is shown in Appendix A, which includes a more detailed process, criteria for judgment, and example analyses.

## 3 CHASM

This section presents the construction and annotation of CHASM, a first-of-its-kind manually curated dataset for detecting covert advertisements on social media. We detail the data collection, human annotation, illustrated in Figure 2. A summary of our dataset statistics is shown in Table 1, with detailed distribution characteristics provided in Appendix G.

## 3.1 Data Collection

**Source Data** Our source data comes from RedNote (also known as Xiaohongshu or RED), a major social media platform in China that has recently gained a growing international user base [29]. The platform mainly hosts content like product recommendations, travel tips, and lifestyle posts. Given its broad influence and frequent mentions of products and paid services, detecting covert advertising in this context is both important and challenging.

Specifically, CHASM was collected using the following three-step pipeline:

**(1) Raw Data Collection:** To eliminate the influence of users' historical behavior on data collection results, we employed three annotators to collect publicly available content from three brand-new accounts with no browsing history. The collected content includes titles, main text, images, comments, and publication dates. The data was collected between September and October 2024. The scope of collection strictly adhered to RedNote's User Privacy Policy. We do not collect any personally identifiable or privacy-sensitive information, such as usernames or IP addresses.

**(2) Data Filtering:** We removed samples with explicit advertising labels, i.e., those marked with sponsored tags, as they are clearly distinguishable from regular content and unlikely to mislead users. These traditional advertisements fall outside the scope of covert advertisements and were excluded from our dataset.

**(3) Data Anonymization:** To further protect user privacy and mitigate the risk of information leakage, we applied anonymization to the dataset using open-source anonymization tools [30, 31]. Specifically, we masked personal information such as names, phone numbers, and email addresses in the text, and obscured potentially privacy-sensitive facial regions in images; examples are shown in Appendix F. We also manually reviewed a random sample of 30 data points after anonymization and found no signs of residual privacy leakage.

## 3.2 Data Annotation

We adopted manual annotation to curate a high-quality dataset. Five native Chinese-speaking students participated as annotators. They were paid $5 per hour, which exceeds the local minimum wage standard. All of them had substantial experience (> 1 hour/day) with RedNote.

Table 1: Statistical Overview of CHASM, containing 4,992 manually high-quality annotated multimodal posts from RedNote. Product-sharing samples refer to posts that mention products but do not have advertising intent. They represent a challenging subset within the non-advertisement samples (see Section 2.2 for further information).

| CHASM Dataset | |
|---|---|
| **Samples** | |
| # Samples | 4992 |
| # Covert Advertisement (Positive) | 612 (12.3%) |
| # Non-Covert Samples (Negative) | 4,380 (87.7%) |
| # Product-Sharing Samples | 1127 (22.6%) |
| **Distribution** | |
| Avg. Images per Sample | 5.28 |
| Avg. Post Text Length | 196.63 |
| Avg. Comments Text Length | 25.01 |
| Time of Earliest Post | Mar. 2020 |
| Time of Latest Post | Oct. 2024 |
| Median Posting Time | Sep. 2024 |
| **Annotation** | |
| # Annotators | 5 |
| Annotations per Sample | 1 - 3 |
| # Annotations | 6474 |
| Avg. annotations per question | 1.30 |
| **Quality Control** | |
| # Test Gold Questions | 50 |
| Accuracy on Gold Questions | 0.94 |

Because of the subjectivity and challenges inherent to the task, and our relatively limited annotation budget, we adopted the following strategies to improve dataset quality and enhance consistency:

**(1) Systematic Annotation Guideline:** We developed systematic, evidence-based, and detailed annotation guidelines to train annotators, accompanied by various examples and analyses. The full guidelines are provided in Appendix A. The annotation interface is shown in Appendix F.

**(2) Gold-Standard Test Questions:** We prepared 70 manually curated gold-standard test questions, designed to be representative and challenging. Each question was discussed among the authors and finalized through group discussion. Among them, 20 questions were used as a qualification test after

Table 2: Zero-shot and in-context learning evaluation results on CHASM. From top to bottom, the two groups are: open-source MLLMs and proprietary MLLMs. **Bold** indicates the best overall performance across all models, and underlined indicates the best within each group. Bold and underlined together indicate that a model is both the best overall and the best within its group. The models marked with an * are reasoning MLLMs. Although GPT-4o and DeepSeek-V3 demonstrate similarly top F1-score performance among all models, **none** of the models are sufficiently reliable for detecting covert advertisements.

| | Metric / Model | Zero-Shot | | | | In-Context Learning | | | |
|---|---|---|---|---|---|---|---|---|---|
| | | P ↑ | R ↑ | F1 ↑ | AUC ↑ | P ↑ | R ↑ | F1 ↑ | AUC ↑ |
| Open | InternVL2.5 | 0.289 | 0.662 | 0.403 | 0.717 | 0.232 | 0.494 | 0.316 | 0.640 |
| | Llava | 0.182 | 0.359 | 0.242 | 0.567 | 0.145 | 0.721 | 0.241 | 0.568 |
| | Qwen2.5-7B | 0.473 | 0.378 | 0.421 | 0.660 | 0.505 | 0.380 | 0.434 | 0.664 |
| | DeepSeek-VL2 | 0.166 | 0.749 | 0.272 | 0.612 | 0.000 | 0.000 | 0.000 | 0.500 |
| | DeepSeek-V3 | **0.499** | 0.787 | 0.571 | 0.826 | **0.578** | 0.607 | **0.592** | 0.772 |
| | Llama-4 | 0.382 | 0.770 | 0.511 | 0.798 | 0.408 | 0.508 | 0.453 | 0.703 |
| Proprietary | Qwen-Max | 0.426 | 0.852 | 0.568 | 0.846 | 0.440 | 0.836 | 0.576 | 0.844 |
| | GLM4-Flash | 0.408 | 0.489 | 0.445 | 0.695 | 0.218 | 0.408 | 0.284 | 0.603 |
| | GLM4-Plus | 0.385 | 0.328 | 0.354 | 0.627 | 0.167 | 0.200 | 0.182 | 0.531 |
| | GPT-4o | 0.464 | 0.836 | **0.597** | **0.851** | 0.442 | 0.633 | 0.521 | 0.762 |
| | GPT-4o-mini | 0.284 | 0.820 | 0.422 | 0.766 | 0.274 | 0.767 | 0.403 | 0.743 |
| | Gemini 2.0 | 0.362 | 0.842 | 0.506 | 0.818 | 0.329 | 0.671 | 0.436 | 0.738 |
| | Step-R1-V-Mini* | 0.455 | 0.750 | 0.566 | 0.813 | 0.444 | 0.721 | 0.550 | 0.798 |
| | QvQ-Max* | 0.485 | 0.402 | 0.440 | 0.631 | 0.244 | 0.836 | 0.378 | 0.737 |
| | Gemini 2.5 Pro* | 0.273 | **0.921** | 0.422 | 0.791 | 0.364 | **0.984** | 0.531 | **0.872** |

annotator training. Annotators were allowed to retake the test multiple times and were required to achieve at least 95% accuracy before beginning formal annotation. The remaining 50 questions were randomly and covertly embedded into the annotation workflow to monitor annotation quality.

**(3) Dynamic Quality Control Strategy:** To improve annotation accuracy while controlling annotation costs, we adopted a dynamic labeling strategy based on the difficulty of each sample. Specifically, for each instance, the first annotator determined whether the content was related to a product or service. If deemed unrelated, the sample was directly labeled as non-covert advertisement. For product-sharing samples, which involve greater subjectivity, we employed a majority voting scheme among three annotators, ensuring that at least one experienced annotator participated. This approach significantly improved annotation quality: the initial inter-annotator agreement (Fleiss' kappa = 0.65) indicates moderate consistency, reflecting the inherent ambiguity of the task. After adopting the dynamic annotation workflow described above, the annotation quality improved markedly: the accuracy on gold-standard questions increased from 78% under single-annotator labeling to 94%, while the required annotation resources were reduced to 43.3% of those needed for exhaustive three-person voting. These results demonstrate that our annotation protocol effectively balances reliability and efficiency in handling this challenging task.

## 4 Evaluation

In this section, we first present the experimental setup. In Section 4.2, we discuss the performance of different MLLMs on the CHASM. Finally, we conduct comparative experiments to investigate which parts of the posts are most helpful for detecting covert advertisements.

### 4.1 Experimental Settings and Metrics

To establish the baseline performance in CHASM, we experiment with 15 different mainstream MLLMs with Chinese language capabilities. We categorize these MLLMs into two groups: open-source MLLMs (containing small- and large-scale models and proprietary MLLMs. Small-scale open-source MLLMs include `Deepseek-vl2-small` [32], `InternVL2.5-8B` [33], `LLaVA-NeXT-8B-hf` [22], `Qwen2.5-VL-7B-Instruct` [23]. Large-scale open-source MLLMs include `Llama-4-Maverick` [34] and `Deepseek-V3` [21]. Proprietary MLLMs include `Qwen2.5-Max`

[35], GLM models [36]: `GLM-4-Flash` and `GLM-4-Plus`, GPT models: `GPT-4o-0806` and `GPT-4o-mini-0718` [20], and `Gemini-2.0-flash` [37]. To evaluate whether reasoning MLLMs can achieve better performance on the covert advertisement detection task, we also include three proprietary reasoning MLLMs: `QvQ-Max` [38], `Gemini 2.5 Pro` [24], `Step-R1-V-Mini` [39].

We consider three different strategies, **Zero-shot Prompting**: The LLM is prompted with a brief judgment criterion along with the full content of the social media post as input, and directly outputs a binary classification indicating whether the content is identified as a covert advertisement; **In-Context Learning**: In addition to using the same input as in zero-shot prompting and the same output format, examples of both labels are additionally provided; **Fine-Tuning**: The same input-output format as zero-shot prompting, and fine-tuned the model using a 5-fold cross-validation setup for prediction. We also provide some additional experimental results for simpler baselines in Appendix C.

We report `Precision`, `Recall`, `F1-Score`, and `AUC`, four standard metrics that respectively assess prediction accuracy, completeness, their balance, and overall classification quality. Considering the imbalance in the distribution of sample labels and our greater emphasis on distinguishing positive examples, we regard the `F1-Score` as the most representative metric. Implementation details of all models, and the training and inference hyperparameters, can be found in Appendix B. The prompt templates are provided in Appendix D.

## 4.2 Main Results

Table 2 shows all models' zero-shot and in-context learning performance. We then fine-tuned the two best-performing small-scale open-source models, and the results are reported in Table 3.

Overall, GPT-4o achieved the highest F1 score in the zero-shot setting, while DeepSeek-V3 performed best with in-context learning. Despite some models showing high recall, precision remained low across both settings. Large-scale open-source MLLMs achieved performance comparable to that of proprietary MLLMs, while both of them outperformed small-scale open-source MLLMs. Among small-scale open-source models, InternVL and Qwen2.5-7B perform better than others.

However, even the top-performing models, GPT-4o and DeepSeek-V3, are **not** sufficiently reliable for detecting covert advertisements, especially regarding the most concerned metric, F1-score; Their best performances are only 0.597

Table 3: Fine-tuning results on CHASM, results show that both models improved statistically significantly (p < 0.01) over zero-shot performance, with Qwen2.5-7B surpassing GPT-4o after fine-tuning, highlighting the effectiveness of our dataset.

| Metric Model | P ↑ | R ↑ | F1 ↑ | AUC ↑ |
|---|---|---|---|---|
| InternVL | 0.681 | 0.520 | 0.590 | 0.743 |
| Qwen2.5 | 0.783 | 0.732 | 0.756 | 0.852 |
| GPT-4o (ZS) | 0.464 | 0.836 | 0.597 | 0.851 |
| Qwen2.5 (ZS) | 0.473 | 0.378 | 0.421 | 0.660 |
| InternVL(ZS) | 0.289 | 0.662 | 0.403 | 0.717 |

and 0.592, respectively. The results empirically show the inherent complexity and subtlety of covert advertisements, indicating that it is challenging for MLLMs to grasp the fine-grained human standards for identifying covert advertisements through prompting alone.

Reasoning MLLMs, such as Step-R1-V-Mini and Gemini 2.5 Pro, achieve relatively good performance in both zero-shot and in-context learning settings. However, their performance does not significantly surpass that of non-reasoning models, particularly in terms of F1-score, where both fall slightly below GPT-4o's zero-shot result. Given their currently higher cost, we argue that reasoning MLLMs do not offer a clear advantage for the covert advertisement detection task at this stage.

Our further analysis shows that in-context learning remains insufficient for our task. Only a few models achieved better performance compared to their zero-shot setting version, which highlights the limitations of in-context learning for this task. We also attempted to include more detailed evaluation criteria in the prompt, as shown in Appendix E, but it did not improve performance.

Table 3 shows the results of fine-tuning the two best-performing open-source models, InternVL and Qwen2.5. The results show that both models improved significantly over their zero-shot performance, with Qwen2.5 achieving superior results. After fine-tuning, Qwen2.5 surpassed the previously best-performing MLLM, GPT-4o, particularly in precision and F1-score. This suggests that fine-tuning effectively equips models to better align with human judgment in identifying covert advertisements.

Conversely, MLLMs under zero-shot settings frequently misclassify normal posts as covert advertisements, resulting in lower precision. These findings underscore the high effectiveness of our dataset in enhancing covert advertisement detection. More detailed error analysis is in Section 5.1.

## 4.3 Which parts of posts help detect covert advertisements?

We utilize the best-performing model, the fine-tuned Qwen2.5-7B, for our experiments. We retrained the model using the same hyperparameters in the absence of images or comments. Our results, shown in Figure 3, indicate that removing either images or comments significantly degrades the model's performance, highlighting that covert advertisement detection is a multi-modal task, and comments also play a critical role in enabling accurate detection.

Figure 3: Impact of Removing Different Modalities on CHASM. Removing either images or comments significantly degrades model performance.

## 5 Discussion

In this section, we provide an in-depth error analysis of CHASM based on more fine-grained human feedback, and pose the following research questions to offer insights for future work.

## 5.1 What types of errors can MLLMs make on CHASM

We analyze the error cases of MLLMs by using fine-grained human feedback to identify common types of mistakes. Specifically, we conducted group discussions to determine the reasons why humans made opposing judgments on a given error case, and categorized them into four distinct error types:

**Insufficient Evidence**: The MLLM misclassified regular posts as covert advertisements without sufficient evidence. These posts typically did not include essential promotional elements and merely mentioned certain brand names.

**Missing Clue**: The MLLM failed to identify clues embedded in the image or comment section, such as shopping links in the comments or requests for private messages for more information.

**Textual Style**: Humans made judgments opposite to the MLLM based on the textual style. E.g., advertisements often employ exaggerated language or use clickbait-style content to attract attention, whereas non-advertisements tend to use a more objective tone.

**Structural Pattern**: The MLLM failed to capture structural features of the post, e.g., recommending products from multiple different brands instead of focusing on a single brand.

We selected the top F1-score models under each evaluation setting: GPT-4o (Zero-shot), DeepSeek-V3 (In-context Learning), and Qwen2.5-7B (Fine-tuned). To enable the comparison, we also included the performance of Qwen2.5-7B before fine-tuning. The results are shown in Table 4. Appendix H shows specific examples of each error type.

We observe a clear divergence in the error distributions when comparing zero-shot or in-context learning approaches to fine-tuned model settings: Fine-tuned MLLMs significantly reduce the misclassification of posts lacking sufficient cues as covert advertisements. This leads to an improvement in precision, thereby enhancing the overall F1-score. In contrast, models like GPT-4o and DeepSeek-V3 often classify posts as covert advertisements even in the absence of clear evidence, including cases where the content is unrelated to any product. Such errors can raise concerns about the reliability of the platform's moderation mechanisms. Therefore, we advocate for using fine-tuned open-source MLLMs, such as Qwen2.5-7B, as a more cost-effective and reliable alternative.

Although the fine-tuned Qwen2.5-7B model demonstrates a decrease in the number of errors across each error category, the results still suggest that there is room for improvement in capturing the structural differences between covert advertisements and non-advertising posts, as well as in identifying subtle cues that may remain in the comment section. We hope these findings offer valuable insights for future model training.

Table 4: Error counts and percentages across the four main categories of error causes in four MLLMs. We selected the top F1-score models: GPT-4o (Zero-shot), DeepSeek-V3 (In-context Learning), and Qwen2.5-7B (Zero-shot and Fine-tuned).

| Error Type | GPT4o(ZS) | DeepSeek-V3(ICL) | Qwen2.5(ZS) | Qwen2.5(FT) |
|---|---|---|---|---|
| **Insufficient Evidence (Total)** | 22 (47.8%) | 16 (36.4%) | 38 (38.8%) | 6 (17.6%) |
| - Misjudged Product Post | 16 (34.8%) | 11 (25.0%) | 30 (30.6%) | 6 (17.6%) |
| - Misjudged Non-Product Post | 6 (13.0%) | 5 (11.4%) | 8 (8.2%) | 0 (0.0%) |
| **Missing Clue (Total)** | 10 (21.7%) | 15 (34.1%) | 32 (32.7%) | 14 (41.2%) |
| - Missed comment clue | 8 (17.4%) | 14 (31.8%) | 26 (26.5%) | 12 (35.3%) |
| - Missed image clue | 2 (4.3%) | 1 (2.3%) | 6 (6.1%) | 2 (5.9%) |
| **Language Style** | 8 (17.4%) | 9 (20.5%) | 16 (16.3%) | 5 (14.7%) |
| **Post Structure** | 3 (6.5%) | 2 (4.5%) | 6 (6.1%) | 6 (17.6%) |
| **Other Errors** | 3 (6.5%) | 2 (4.5%) | 6 (6.1%) | 3 (8.8%) |

## 5.2 Research Directions For Further Investigation

Due to limitations in data availability, we were unable to incorporate certain features into our study, which made it difficult to identify covert advertisements in some cases. We advocate that social media moderators consider the following strategies to improve detection accuracy:

**Dynamics Detection**: We argue that the labeling of covert advertisements is not static, but evolves along with the post's dynamics in the comment section. Therefore, unlike other social media moderation tasks, our task should be designed with a greater emphasis on temporal sensitivity, rather than relying solely on labeling at the time of posting. We thus encourage future work to consider the dynamic nature of covert advertisements in detection frameworks.

**User Behavior Data**: User feedback data is crucial for detecting covert advertisements, as it reflects users' satisfaction and reactions to the content. Due to limitations in data accessibility, we were unable to analyze this aspect in our study. However, we believe that social media platforms could consider incorporating user behavior signals, such as likes, viewing duration, and report frequency, into a more comprehensive framework for identifying covert advertisements.

**Creator Profiling**: Historical data on content creators can be useful for detecting soft advertisements. For example, inconsistencies between a post's style or topic and a user's previous posts or the user's historical credibility may serve as important signals. Due to privacy concerns, we did not collect any user-related information in this study. Future research could explore the integration of creator-level features into detection frameworks.

## 6 Related Work

**ML for Social Media Content Moderation**    Moderating social media content is crucial for ensuring fair business practices, maintaining social order, and safeguarding mental health [40]. Current research focuses on identifying various types of harmful content, including hate speech [18], fake news [14], rumors [41], cyberbullying [16], toxic content, and child abuse material [17]. Recently, detecting machine-generated text has also emerged as a critical task, given the increasing use of AI-generated content to manipulate public opinion [42, 43]. Specifically, hate speech detection often combines text analysis with social network analysis [44], while fake news detection involves verifying the authenticity of news by comparing similar content [14]. Rumor and cyberbullying detection, on the other hand, predominantly leverage NLP methods to analyze textual data [45, 46]. While existing work addresses various forms of harmful content, much of it is either hard to conceal or can be verified using objective references, such as in fact-checking. Covert advertisements, however, are

deliberately subtle and deceptive, making their detection more challenging and demanding additional effort.

**Advertisement Dataset**   Existing related datasets focus on traditional advertisements, such as [47] collected 20K official Facebook ads to predict revenue, and [48] compiled 64K advertisement images and 3K videos. Similarly, [49] gathered 1K advertisement images to analyze user visual attention, and [50] collected 48K textual Chinese advertisement posts to assess legality. These datasets were not collected for advertisement detection, but rather for conducting further analysis on advertisements. [51] introduces a dataset for advertising, but its data collection is biased, and the task formulation is insufficiently aligned with real-world covert advertising behaviors. In contrast, covert advertising content is inherently highly deceptive and concealed, which is why our task primarily focuses on identifying such content.

# 7   Conclusion

In conclusion, this study introduces CHASM, the first dataset designed to evaluate the capabilities of LLMs for detecting covert advertisements on social media. Our evaluations indicate that covert advertisements are inherently deceptive, and current MLLMs are not sufficiently reliable in detecting them without additional training. Given these challenges, our dataset offers a valuable foundation for fine-tuning open-source MLLMs, enabling notable improvements in their ability to detect covert advertisements. The error analysis highlights key areas for further enhancement, such as detecting structural differences in posts and uncovering highly subtle advertising cues. We hope our work serves as a call to raise awareness of covert advertisements on social media and to encourage improvements in MLLMs to help maintain a more honest and fair social media environment.

# 8   Limitation

Our research is limited to the Chinese internet platform RedNote. Although it is one of the most influential commodity-sharing-centered social media platforms in the world, we still advocate for extending covert advertisement detection to a broader range of domains. In China, the discussion could also include other social media platforms such as Douyin[5] and Weibo[6]. At the same time, we believe that covert advertisement detection can be expanded to support multiple languages, serving people worldwide. Due to limitations in human resources, we did not construct a larger and more comprehensive dataset. We encourage future work to build datasets that are both larger in scale and broader in coverage. For constraints in data availability, our dataset does not incorporate more comprehensive user behavior information, which we believe could play an important role in improving covert advertisement detection.

## Acknowledgments

This work is partially funded by the Guangdong Provincial Key Lab of Integrated Communication, Sensing, and Computation for Ubiquitous Internet of Things (No. 2023B1212010007) and Yangcheng Scholars Research Project (No.2024312049). We also thank the Pioneer Centre for AI for funding Tianyi Hu's PhD, and the Department of Computer Science at Aarhus University for providing his travel support.

---

[5]https://www.douyin.com/
[6]https://weibo.com/

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

# A    Guideline of Detecting Covert Advertisement

**Observation object:**    In order to effectively evaluate whether a post is a hidden advertisement, the annotator should pay comprehensive attention to all parts of the post. Specifically, the annotator needs to focus on the image, body content, and comments

**Identify content:**    The annotator should first determine whether the content is related to a product or paid service. If it clearly falls into a category unrelated to commercial goods, it can be simply classified as non-advertising content (**Option 1**). The annotator's next task is to determine whether the content is a covert advertisement. It is important to avoid misclassifying general lifestyle sharing content as advertising. Annotators should carefully distinguish between the two based on the following evidence:

Table 5: Common Evidence of Covert Advertising in Social Media Content

| **Common Characteristics of Covert Advertisements** |
| --- |
| 1. Often include detailed product information such as price, purchase method, and product address. |
| 2. Frequently contain purchase links, either embedded in the image or placed in the comment section. |
| 3. May direct followers to join groups, message privately, or move to external platforms. |
| 4. Comment sections may include remarks from users pointing out that the content is an ad. |
| 5. Often use irrelevant but popular product tags to attract unrelated traffic. |
| 6. Commonly promote unknown products or counterfeit versions of well-known items. |
| 7. May use clickbait-style or eye-catching titles to draw attention. |
| 8. Tend to focus on a single product or a set of products from the same brand, rather than covering diverse items. |
| 9. Adopt formal or commercial-style language, while lifestyle content tends to be casual and personal. |
| 10. Rarely mention disadvantages; instead, ads often exaggerate product strengths. |
| 11. Use exaggerated promotional phrases, such as "best of the year" or "unbeatable value." |
| 12. Brand names appear repeatedly and are visually emphasized in both text and images. |
| 13. The product is usually the central focus, unlike non-advertising content that may highlight other themes like travel or personal experiences. |

**Typical examples:**    We have summarized several common types of covert advertisements for the annotator's reference. Covert advertisements can take various forms, such as images displaying the name of the online shop and product, or comments explicitly mentioning the shop name. In some cases, comments may subtly convey product or shop names in complex ways, or images and comments may include product descriptions that hint at where to find the link. Other examples include text making clear references to a product, comments suggesting private messages to share product links, product names visible directly in the image, or even product links hidden in flipped or reversed images. These examples serve as a guide but do not cover all possible manifestations of covert advertisements. We show some typical examples in Figure 4.

# B    Implement Details

The details of the models, including their parameter sizes and download links, are summarized in Table 6.

In our setup, we fine-tuned the model by inserting LoRA adapters (rank 8, $\alpha = 32$) into all linear layers, using micro-batches of size 1 with gradient accumulation over 16 steps to emulate a larger effective batch. Optimization was handled by AdamW ($\beta_1 = 0.9$, $\beta_2 = 0.95$, $\epsilon = 1 \times 10^{-8}$) at a learning rate of $1 \times 10^{-4}$ with a weight decay of 0.1, guided by a cosine scheduler (no warmup) across three epochs. Inputs were truncated to 4096 tokens using the `delete` strategy, and `bfloat16` mixed precision was enabled to improve speed and reduce memory usage.

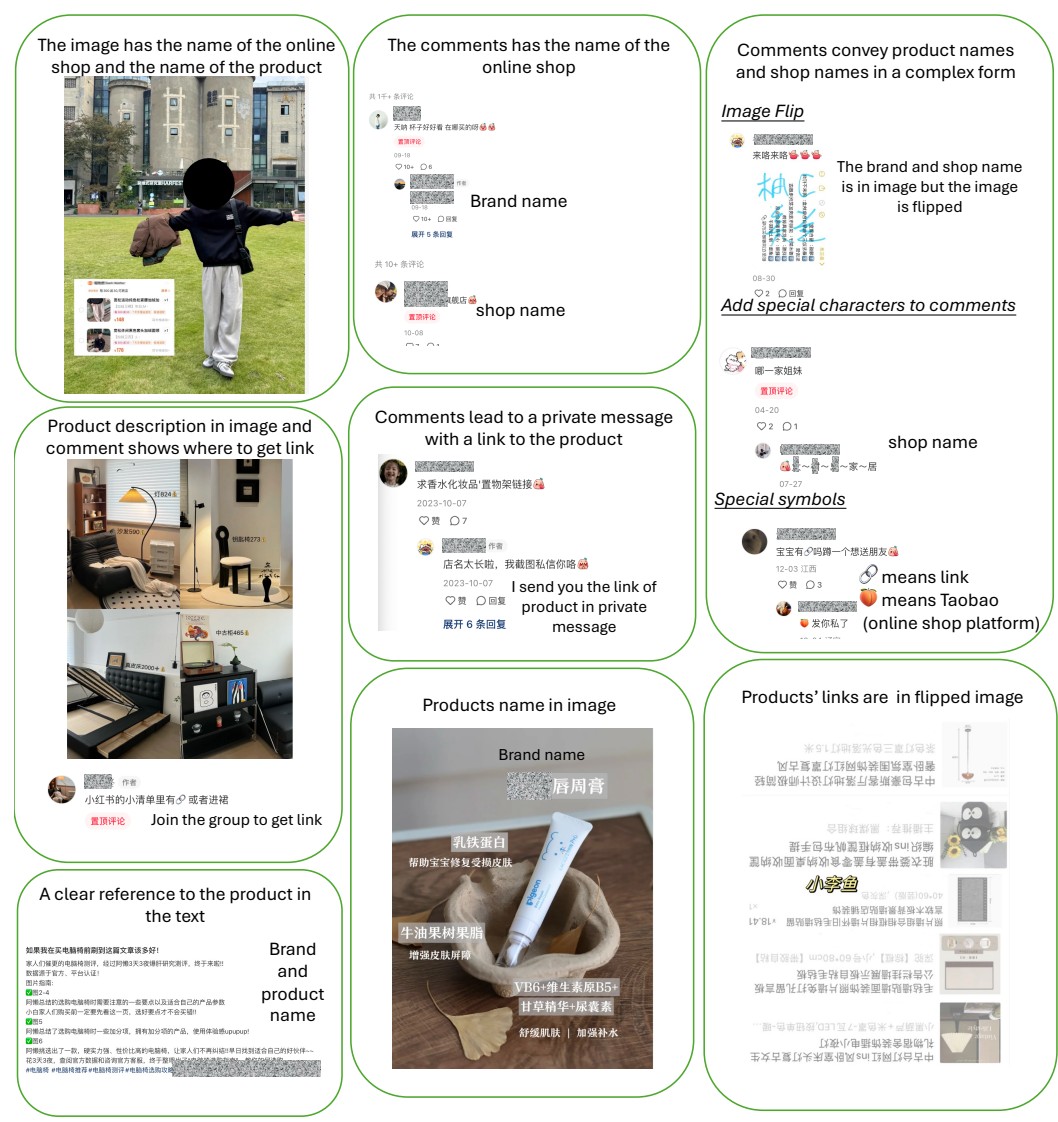

Figure 4: Typical Examples of Covert Advertisements

# C Supplementary experimental results

We trained and evaluated several lightweight models on the CHASM dataset. The results are shown in Table 7. These baselines underperform compared to fine-tuned MLLMs, further highlighting the difficulty and subtlety of the task.

Table 6: Open-source and proprietary MLLMs with parameter counts and links.

| Model | Parameters | Link |
|---|---|---|
| Deepseek-vl2-small [32] | 16B | Model_Link |
| InternVL2.5-8B [33] | 8B | Model_Link |
| LLaVA-NeXT-8B-hf [22] | 8B | Model_Link |
| Qwen2.5-VL-7B-Instruct [23] | 7B | Model_Link |
| Deepseek-V3 [21] | 671B | Model_Link |
| Llama-4-Maverick [34] | 400B | Model_Link |
| Qwen2.5-Max [35] | - | Model_Link |
| GLM-4-Flash-250414 [36] | - | Model_Link |
| GLM-4-Plus [36] | - | Model_Link |
| Gpt-4o-2024-08-06 [20] | - | Model_Link |
| Gpt-4o-mini-2024-07-18 [20] | - | Model_Link |
| Gemini-2.0-flash [37] | - | Model_Link |
| QvQ-Max [38] | - | Model_Link |
| Step-R1-V-Mini [39] | - | Model_Link |
| Gemini 2.5 Pro [24] | - | Model_Link |

Table 7: Performance of lightweight baseline models on the CHASM dataset.

| Model | Accuracy | Precision | Recall | F1 Score |
|---|---|---|---|---|
| TF-IDF + LR | 0.647 | 0.655 | 0.648 | 0.644 |
| TF-IDF + SVM | 0.624 | 0.629 | 0.624 | 0.620 |
| BERT + ResNet | 0.653 | 0.722 | 0.653 | 0.615 |
| CLIP | 0.622 | 0.622 | 0.622 | 0.622 |

# D  Prompt Template

---

**Zero-shot Prompt**

Your task is to determine whether a social media post contains advertising content. The input may include tweets, images, and comments. If the input contains persuasive content encouraging shopping, output '1' to indicate the presence of an advertisement. If the input is just general life-sharing content or unrelated to products, output '0'. Please output only '1' or '0' without any additional text.

---

**Few-shot Prompt**

Your task is to determine whether a social media post contains advertising content. The input may include tweets, images, and comments. If the input contains persuasive content encouraging shopping, output '1' to indicate the presence of an advertisement. If the input is just general life-sharing content or unrelated to products, output '0'. Please output only '1' or '0' without any additional text.

[A Selected Convert Advertisement Example]

[A Selected Non-Convert Advertisement Example]

---

# E  Can more detailed prompts lead to better detection performance?

Because the experiments in Section 4.2 show that, in both zero-shot and in-context learning settings, MLLMs do not follow the same criteria as humans when identifying covert advertisements, we attempted to provide more detailed evaluation standards directly in the prompt. However, as shown in

Table 8, this did not help align with fine-grained human standards, and these more detailed prompts performed worse.

We use the templates as follows:

---

**Detailed Zero-shot Prompt**

Your task is to determine whether the social media tweets contain advertising content. The input may include tweets, pictures, and comments. If the input contains content that persuades people to buy, the output is '1', which means it contains advertising. If the input is just general life sharing content or other content not related to the product, the output is '0'. Please only output '1'/'0', and do not output other content.

Here are some guidelines: 1. Clear evidence of promotion: Hidden ads often contain obvious signs of promotion, such as providing direct purchase links or product purchase instructions. To make the ads more hidden, promotional links are sometimes embedded in pictures or comments, or users are redirected to private chat groups for sales. In contrast, non-advertising content focuses mainly on sharing personal experiences, so it may only casually mention product or store names, and the content usually lacks enough information for users to complete the purchase. 2. Post language style: Hidden ads often use clickbait-style titles and sales pitches. Such articles often have a strong promotional tone and use exaggerated language to emphasize the advantages of the product, which runs counter to the natural style of daily communication. In contrast, non-advertising content is usually more casual in tone and focuses on sharing personal experiences rather than promoting products. It may also mention product shortcomings. 3. Post text and image structure: Hidden ads often focus text and images on a single specific product or closely related products of the same brand. In contrast, non-promotional lifestyle sharing posts often involve multiple different brands in the same category, some of which may even be competitors, or the author does not explicitly recommend any specific brand.

---

**Detailed Few-shot Prompt**

Your task is to determine whether the social media tweets contain advertising content. The input may include tweets, pictures, and comments. If the input contains content that persuades people to buy, the output is '1', which means it contains advertising. If the input is just general life sharing content or other content not related to the product, the output is '0'. Please only output '1'/'0', and do not output other content.

Here are some guidelines: 1. Clear evidence of promotion: Hidden ads often contain obvious signs of promotion, such as providing direct purchase links or product purchase instructions. To make the ads more hidden, promotional links are sometimes embedded in pictures or comments, or users are redirected to private chat groups for sales. In contrast, non-advertising content focuses mainly on sharing personal experiences, so it may only casually mention product or store names, and the content usually lacks enough information for users to complete the purchase. 2. Post language style: Hidden ads often use clickbait-style titles and sales pitches. Such articles often have a strong promotional tone and use exaggerated language to emphasize the advantages of the product, which runs counter to the natural style of daily communication. In contrast, non-advertising content is usually more casual in tone and focuses on sharing personal experiences rather than promoting products. It may also mention product shortcomings. 3. Post text and image structure: Hidden ads often focus text and images on a single specific product or closely related products of the same brand. In contrast, non-promotional lifestyle sharing posts often involve multiple different brands in the same category, some of which may even be competitors, or the author does not explicitly recommend any specific brand.

[A Selected Convert Advertisement Example]

[A Selected Non-Convert Advertisement Example]

---

Table 8: Evaluation metrics under top performance models and different prompt settings. Compared to the prompts used in the main content (Normal Prompt), we found that using prompts with more detailed evaluation criteria information did not help align with fine-grained human standards; These more detailed prompts performed worse.

| Model | Prompt Type | Precision | Recall | F1-score | AUC-ROC |
|---|---|---|---|---|---|
| GPT-4o (ZS) | Detailed Prompt | **0.482** | 0.672 | 0.562 | 0.786 |
| | Normal Prompt | 0.464 | **0.836** | **0.596** | **0.851** |
| DeepSeek-VL3 (ICL) | Detailed Prompt | 0.565 | 0.574 | 0.569 | 0.756 |
| | Normal Prompt | **0.578** | **0.607** | **0.592** | **0.772** |

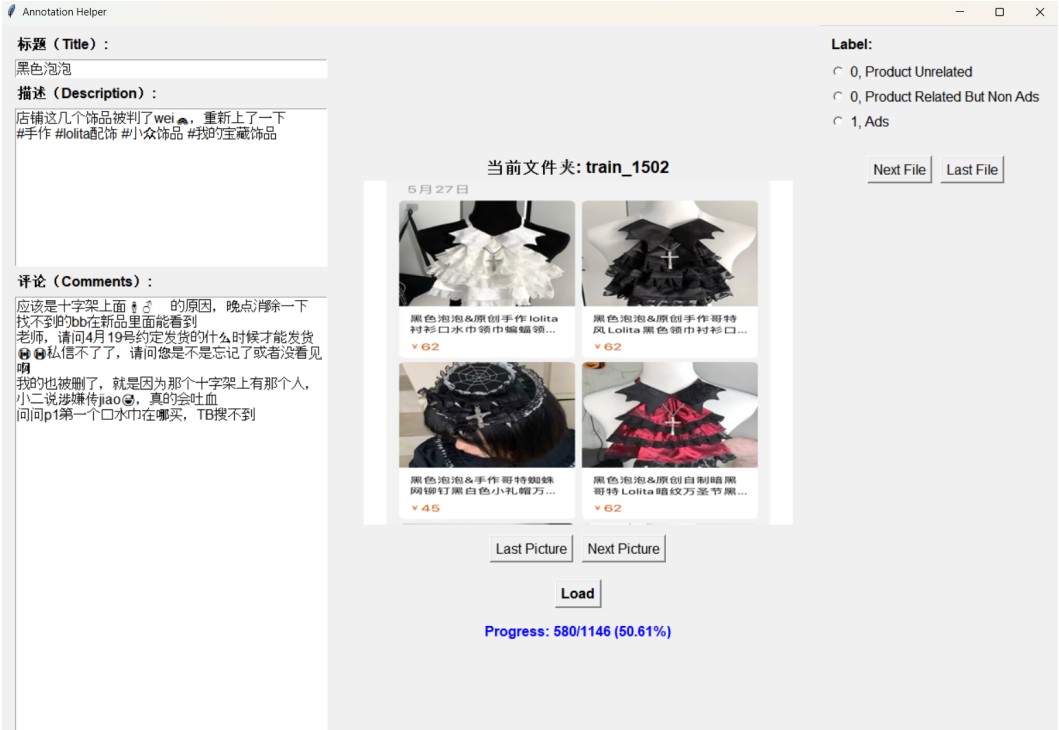

Figure 5: Screenshot of The Annotation System

# F  Demos of CHASM

## F.1  Screenshot of The Annotation System

Figure 5 shows the annotation interface designed for labeling social media posts. Title, Description, and Comments fields on the left, displaying the textual content of the post. A preview of associated images in the center, A labeling section on the right, where annotators can choose from three options: Product Unrelated, Product Related But Non-Advertisement, and Covert Advertisement.

## F.2  Examples of Anonymization

### F.2.1  Examples of Text Anonymization

In the examples, we masked detailed information such as detailed addresses or the website.

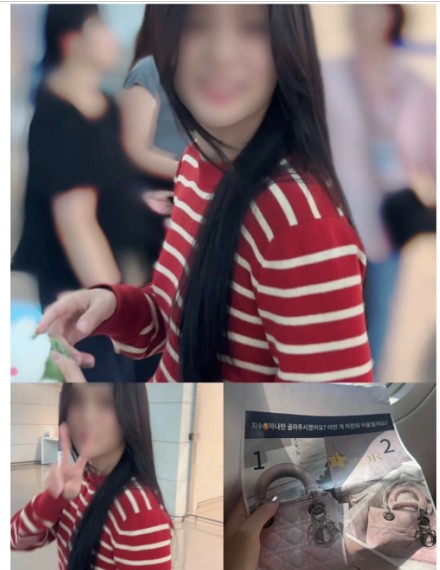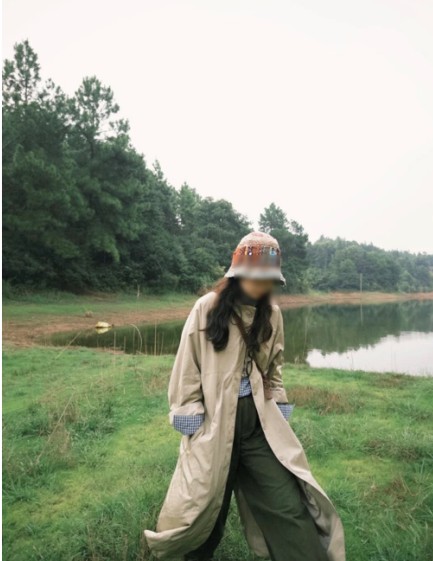

Figure 6: Example of the image anonymization

<div>

**Example 1**

Chinese Text: <详细地址>的某华公寓，后面就是工业园，超级吵白天晚上都吵

Translate: The Mouhua Apartment at <detailed address> is right next to an industrial park. It's extremely noisy both during the day and at night.

</div>

<div>

**Example 2**

Chinese Text:虽然，但是文件要自己命名和管理才知道是什么，在哪里。ai代理的话我怎么找到呢? <网址>

Translate: Although... the files need to be named and organized manually, so I know what they are and where they are. If it's handled by an AI agent, how would I be able to find them? <website>

</div>

### F.2.2 Examples of Image Anonymization

As shown in Figure 6, we anonymized the images, primarily by masking faces, to further protect privacy.

## G Distribution of the Dataset

This section illustrates how normal posts and covert advertisements differ in their distributions over five key features, shown in Figure 7. The five key feature dimensions are Number of Images, Post Text Length, Number of Comments, Average Comment Length, and Number of Tags. Blue bars represent the count distribution of normal posts (left y-axis). Red bars represent the count distribution of covert advertisement posts. Blue lines indicate the density of normal posts across the feature values (right y-axis). Red lines indicate the normalized ratio of covert ads across the feature values.

Although there are some distributional differences between the two, for example, covert advertisements tend to have slightly shorter text lengths than normal posts, these statistical features are overall quite similar and are insufficient on their own to reliably distinguish covert advertisements from normal posts.

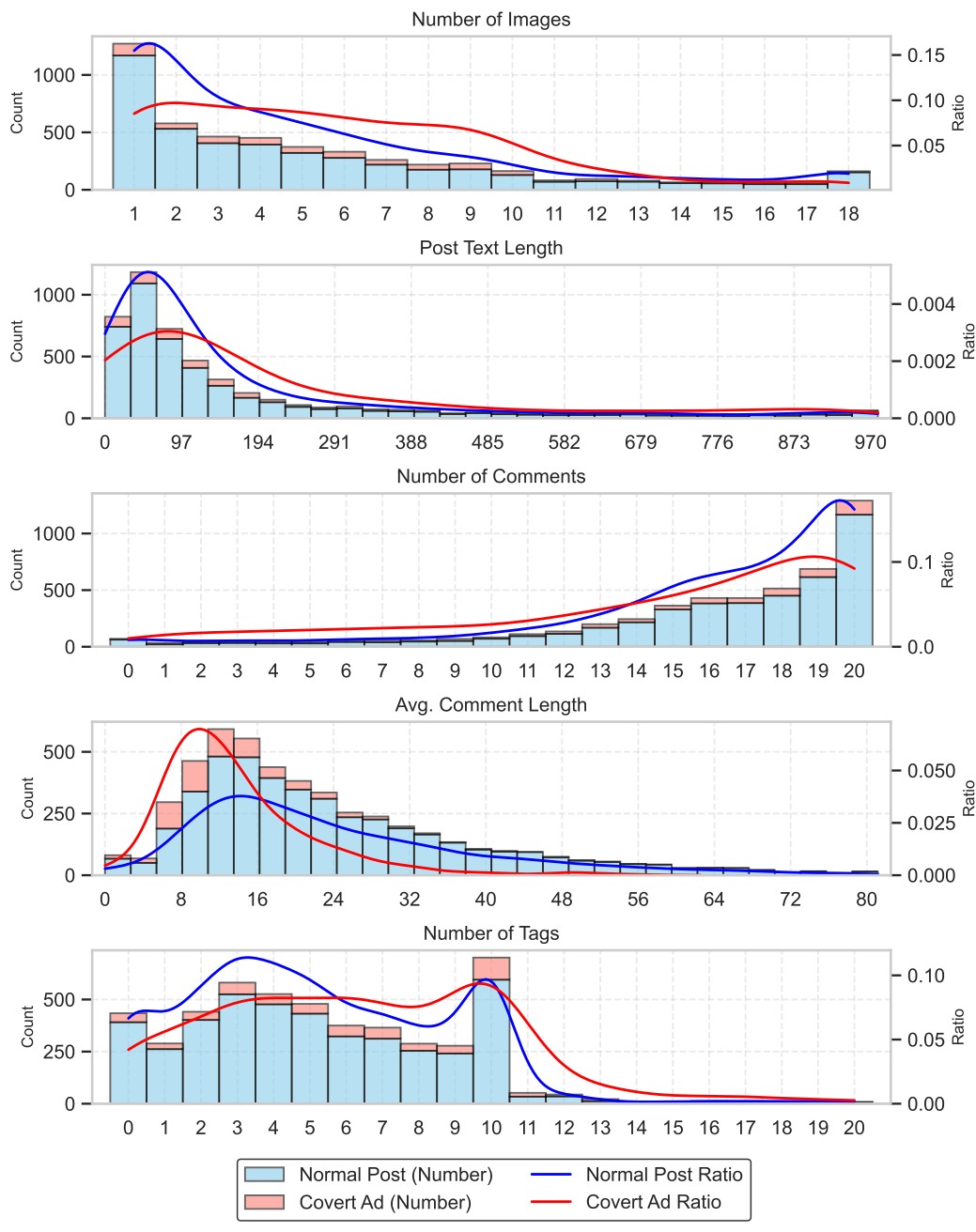

Figure 7: Feature Distributions of Normal Posts and Covert Advertisements

## H   Examples of the Error Types

In this chapter, we discuss concrete examples of the four common error types listed in Section H, as illustrated in Figure 8.

The first common issue is failing to detect hidden clues in the comments or images. As shown in the top row of Figure 8, the left subfigure contains a large highlighted area (red box) showing a specific branded product along with its price, which indicates a clear promotional intent. In the right subfigure, the red box highlights a comment asking users to send a private message, a common tactic used to evade platform review while promoting products.

The second common issue is mistakenly classifying normal posts as advertisements without a factual basis. As shown in the middle row of Figure 8, the left subfigure features a post recommending a novelist. Although the language style may resemble promotional wording, the content itself is unrelated to any product or advertisement and should not be considered an advertisement. The right subfigure shows a post asking for opinions on outfit choices. While it may touch on product-related topics, the author's focus is on seeking advice rather than promoting any specific item.

The third common issue involves structural cues. For example, in the left subfigure of the bottom row in Figure 8, the content introduces multiple skincare products. The structure of the post is centered around summarizing a variety of items rather than focusing on a single one. Since these products are competing within a narrow category, it is less likely that the post serves as an advertisement.

The fourth issue relates to linguistic style cues. For example, in the right subfigure of the bottom row in Figure 8, the post introduces a certain medication. The writing style resembles personal lifestyle sharing, and a significant portion of the text is dedicated to discussing its drawbacks. Therefore, it should be classified as normal sharing content rather than an advertisement.

# I  Broader impacts

Our work has the potential to generate a positive social impact. Covert advertisement is a deceptive practice that seeks to gain unfair competitive advantages and is explicitly prohibited by advertising laws in multiple regions, including China and the United States. By enabling the automatic detection of covert advertisements, we believe our approach can help platforms foster a fairer and more trustworthy social media environment.

The project may also have negative impacts, such as the risk of mistakenly classifying legitimate posts as advertisements, which could lead to an unsatisfactory user experience. However, nearly all quality-control ML models face this kind of issue, so the negative impact is neither significant nor unique to our model. We advocate for a cautious approach in the application of automatic advertisement detection by platform administrators. For instance, any punitive actions against users should involve human review, and platforms should provide clear channels for user feedback and explanation to ensure that the normal user experience is not adversely affected.

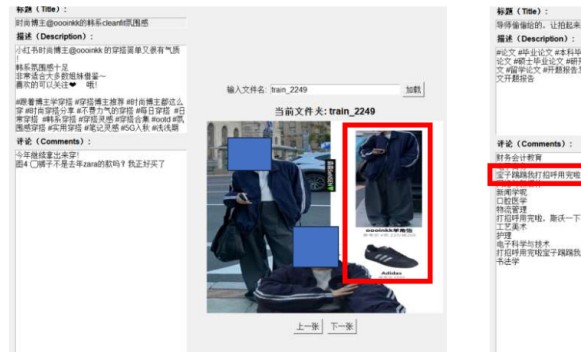
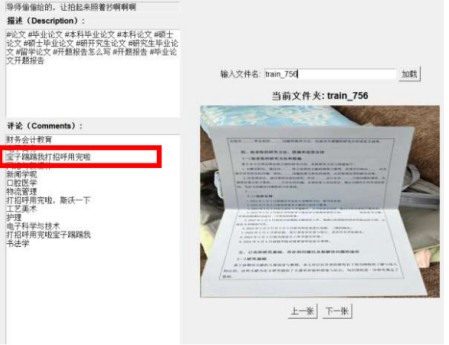

Predict: 0 Label : 1
Error Type: **Missing Clue in Image**

Predict: 0 Label : 1
Error Type: **Missing Clue in Comment**

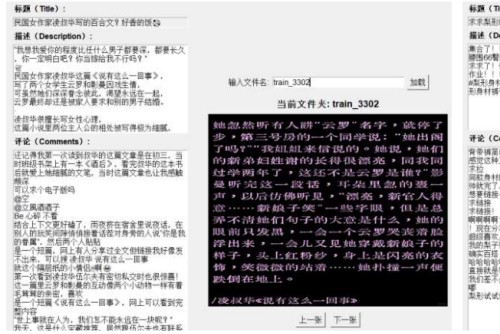
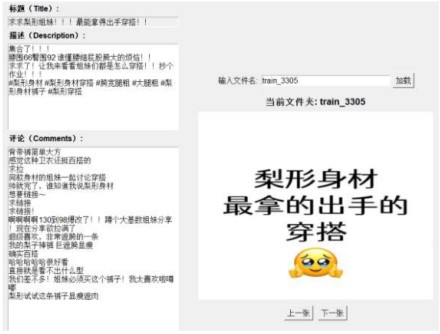

Predict: 1 Label : 0
Error Type: **Misjudged Non-Product Post**

Predict: 1 Label : 0
Error Type: **Misjudged Product Post**

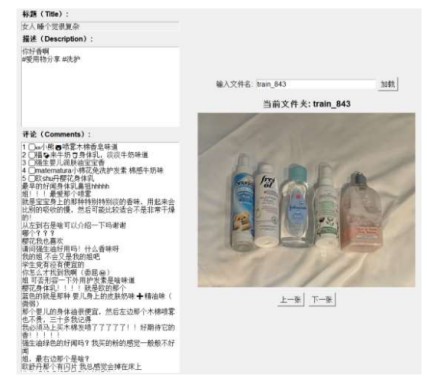
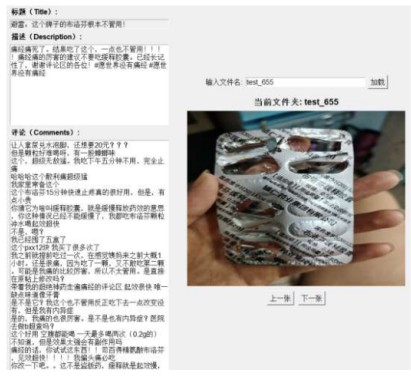

Predict: 1 Label : 0
Error Type: **Post Structure**

Predict: 1 Label : 0
Error Type: **Language Style**

Figure 8: Examples of Six Different Error Types

