# OpenReview forum: "CHASM: Unveiling Covert Advertisements on Chinese Social Media"
_NeurIPS.cc/2025/Datasets_and_Benchmarks_Track — NeurIPS 2025 Datasets and Benchmarks Track poster_

### Official Review · Reviewer_b1kH · 2025-06-03

**Rating:** 5
**Confidence:** 3

**Summary:**

This paper proposes a new task of detecting covert ads. Based on real-world examples from RedNote, authors introduce CHASM, a new dataset for evaluating the ability of MLLMs to detect covert ads. Besides, they also conduct a comprehensive evaluation of CHASM using various open-source and closed-source MLLMs.

**Additional Feedback:**

- Investigate the impact of text-only vs. multimodal inputs to isolate the value of visual features.
- I want to know how well the fine-tuned pre-trained small model works.  Given the similarity of the tasks, you can consider using the existing multimodal fake news detection model or the clip model to see the effect.
-  It is recommended to refine Table 1. Based on my understanding, this is a binary classification dataset. Does the "Positive Sample" listed in Table 1 refer to the number of non-covert ads, and is "Product-Related Samples" a subset of covert ads? It would be helpful to clearly report the total number of covert ads and non-covert ads. In addition, is the dataset balanced? This information is crucial for its future use.

**Dataset Code Accessibility:**

Yes

**Dataset Code Comments:**

Code accessible.

**Ethical Considerations:**

No, there are no or only very minor ethics concerns

**Ethics Flags:**

["Improper research involving human subjects", "Safety and security", "Human rights (including surveillance)"]

**Final Justification:**

The additional analyses, particularly the inclusion of lightweight baselines, are appreciated and strengthen the empirical contributions of the paper. The clarification regarding cross-platform applicability and the dataset characteristics of RedNote also provides helpful context. While the current number of labeled covert ads (612) provides a useful starting point, I encourage authors to consider expanding its scale in future work. I will maintain my original score.

**Limitations Weaknesses:**

- As discussed in paper, CHASM is rooted in Chinese social media (Rednote), which may limit generalizability to other languages/platforms.
- The experiment was conducted on a variety of large models, and there was no fine-tuned pre-trained small model baseline.
- Some tables need further improvement, like Table 1.

**Strengths Contributions:**

- This paper proposes a novel covert ad detection task, which is of great practical significance, helping platforms create a more trustworthy social media environment, promoting fair competition in business, and improving user experience and protecting user rights.
- it provides a high-quality, privacy-compliant dataset for covert ads detection with rigorous annotation protocols, filling a resource gap.
- Overall, it is well-written and highly engaging.
- The experiments are quite comprehensive and show that the current MLLM is not sufficient to reliably detect covert ad without additional training, providing directions for future work.

---

> ### Author Rebuttal · Authors · 2025-07-29
>
> We sincerely appreciate your valuable time and insightful comments! We hope that our responses below will address your concerns.
>
> **1. Generalizability to other languages/platforms.**
>
> We agree that cross-lingual generalization is important. During the rebuttal phase, we examined public Instagram datasets but found they lacked covert advertisements as defined in our paper. Most ads were overt ads with clear promotional tags (e.g., “#ad”, “#sponsored”) or direct sales content, and thus do not qualify as deliberately disguised.
>
> We also clarify that RedNote is an app originally built around shopping guides, which naturally leads to a higher concentration of product-related and advertorial content. This is in contrast to platforms such as Facebook and Instagram, as mentioned by the reviewer, where the proportion of such content is significantly lower. Therefore, RedNote provides a more abundant and accessible source of relevant samples and represents a more pressing issue to address.  We aim to tackle the challenges of collecting sufficient covert ad data from platforms like Facebook and Instagram in future work.
>
> **2. Some more baseline.**
>
> We appreciate the reviewer’s suggestion. In addition to evaluating large-scale MLLMs, we have now included results from fine-tuned small-scale models as baselines.
>
> Specifically, we trained and evaluated several lightweight models on the CHASM dataset. These include:
>
> | Model         | Accuracy | Precision | Recall | F1 Score |
> | ------------- | -------- | --------- | ------ | -------- |
> | TF-IDF + LR   | 64.77%   | 65.51%    | 64.77% | 64.38%   |
> | TF-IDF + SVM  | 62.40%   | 62.88%    | 62.40% | 62.00%   |
> | BERT + ResNet | 65.34%   | 72.18%    | 65.34% | 61.50%   |
> | CLIP          | 62.20%   | 62.20%    | 62.20% | 62.19%   |
>
> All models were trained on the same data as the MLLMs. These baselines underperform significantly compared to fine-tuned MLLMs, further highlighting the difficulty and subtlety of the task.
>
> We thank the reviewer for pointing this out and have added these baselines to strengthen the empirical comparisons.
>
> **3. Some tables need further improvement, like Table 1.**
>
> We acknowledge that Table 1 and potentially others could be improved for clarity and readability. We will refine the formatting in the camera-ready version.
>
>
> **4.Investigate the impact of text-only vs. multimodal inputs to isolate the value of visual features.**
>
> We conduct a modality ablation study in Section 4.3 (Figure 3) to assess the impact of visual features. The results show that multimodal models significantly outperform text-only and image-only variants, demonstrating that visual cues contribute meaningfully to detecting covert advertisements. This confirms that image information provides complementary signals beyond textual content.
>
> Best regards,
>
> Authors

---

> > ### Comment · Reviewer_b1kH · 2025-08-05
> >
> > Thank you for your thoughtful response. The additional analyses, particularly the inclusion of lightweight baselines, are appreciated and strengthen the empirical contributions of the paper. The clarification regarding cross-platform applicability and the dataset characteristics of RedNote also provides helpful context. However, one point raised in the Additional Feedback remains unanswered, specifically: "Does the 'Positive Sample' listed in Table 1 refer to the number of non-covert ads, and is 'Product-Related Samples' a subset of covert ads? In addition, is the dataset balanced?" This information is essential for understanding the dataset’s composition and its future applicability. Given these reasons, I will maintain my original score.

---

> > > ### Author Response · Authors · 2025-08-05
> > >
> > > Thank you for your response to our clarification. Regarding your question, we are happy to provide further explanation.
> > >
> > > As stated in Section 3.2, in Table 1, the term **Positive Samples** refers to **covert ads**, while **Product-Related Samples** refer to posts that are related to products. **Covert ads** (i.e., *Positive Samples*) form a subset of the **Product-Related Samples**, since covert ads must exhibit clear marketing intent and thus are necessarily product-related. However, not all product-related posts are covert ads: some may simply be regular user-generated content involving product sharing or personal opinions. These represent a more challenging class of negative (non-ad) samples.
> > >
> > > Out of the 4992 total samples, 612 are labeled as covert ads, and the remaining 4380 (4992 - 612) are non-ads. Among these non-ad samples, 515 (1127 - 612) are difficult non-ad samples, which we curated specifically to enhance the challenge and realism of the dataset. As these figures indicate, our dataset is imbalanced, which reflects the real-world distribution of covert advertisements on social media platforms.
> > >
> > > Thank you for your question. We will clarify these definitions more explicitly in the camera-ready version to avoid any potential misunderstanding.
> > >
> > > Best regards,
> > > Authors

---

> ### Comment · Reviewer_b1kH · 2025-08-06
>
> Thank you for the clear and helpful clarification. I appreciate your efforts in curating challenging negative samples and in making the dataset reflective of real-world distributions. While the current number of labeled covert ads (612) provides a useful starting point, I encourage you to consider expanding its scale in future work.

---

### Official Review · Reviewer_QrFi · 2025-06-10

**Rating:** 5
**Confidence:** 4

**Summary:**

This paper introduces CHASM, a novel dataset for evaluating how well multimodal large language models (MLLMs) can detect covert advertisements on Chinese social media. Covert ads deliberately disguise themselves as regular posts to influence purchasing decisions without disclosing their promotional nature. The authors collected 4,992 posts from the RedNote platform and developed systematic annotation guidelines for identifying covert advertisements. Their experimental evaluation showed that current MLLMs perform poorly at this task - even the best model (GPT-4o) achieved only 59.7% F1-score in zero-shot settings. While fine-tuning open-source MLLMs on CHASM improved performance significantly (reaching 75.6% F1-score), the models still struggled with detecting subtle advertising cues and structural patterns. The authors conducted detailed error analysis revealing specific challenges like recognizing visual/textual patterns and identifying embedded promotional content in comments. This work establishes a foundation for improving automated detection of covert advertising while highlighting key areas needing further research.

**Dataset Code Accessibility:**

Yes

**Ethical Considerations:**

No, there are no or only very minor ethics concerns

**Final Justification:**

I have maintained my original score

**Limitations Weaknesses:**

### About the dataset

1. While the annotation guidelines (Appendix A) are detailed, the definition of "covert advertisement" is inherently subjective. The paper does not sufficiently justify how the objectivity and universality of these guidelines were ensured—for instance, whether marketing or legal experts were consulted to validate the definition's soundness—which somewhat weakens the guidelines' authority.
2. The error analysis (Section 5.1) categorizes and quantifies error types across models but lacks analytical depth. For example, the paper notes that fine-tuning significantly reduces "Insufficient Evidence" errors but fails to deeply investigate the root cause: did the model genuinely learn to understand "promotional intent," or did it merely overfit to certain superficial features in the training data?
3. The paper frames the absence of "user behavior data" and "creator profiling" as "future research directions," but these are more accurately significant limitations of the current study and dataset. Without this critical information, a ground-truth label for many ambiguous cases may be fundamentally unattainable. The paper should more directly discuss this as a limitation of the present work.



### About the experiment

1. The experimental section is limited to a single platform (RedNote), and additional experiments on other social media platforms (e.g., Weibo, Douyin) should be included to demonstrate the method’s cross-platform generalization.
2.  The ablation studies only address modality contributions, omitting the detection difficulty of different advertisement types (e.g., comment-based vs. image-based); further ablation studies on advertisement types are recommended.


### About writing

1. Suggest changing the section heading for better academic tone, for example, change "4.1 Experiments Settings and Metrics" to "4.1 Experimental Settings and Metrics".
2.  Line 202: Suggest correcting a grammatical error in a list description, for example, change "open-source MLLMs (contains small-scale and large-scale model)" to "open-source MLLMs (containing small- and large-scale models)".
3.  Line 341: Suggest improving phrasing for clarity and flow, for example, change "Existing datasets in related focuses on traditional advertisements..." to "Existing related datasets focus on traditional advertisements...".
4.  Suggest standardizing preposition usage for consistency, for example, change "...detecting covert advertisements in Social Media." to "...detecting covert advertisements on social media." in the conclusion.

### Additional Questions

1.  In the "difficulty-aware dynamic annotation workflow" for difficult samples requiring a three-annotator vote, what was the initial inter-annotator agreement (e.g., Fleiss' Kappa or Cohen's Kappa) before the group discussion and final decision? This metric would provide a more direct measure of the task's inherent ambiguity.
2.  The results show that fine-tuning significantly outperforms in-context learning, even with detailed prompts (Appendix D). What is your hypothesis as to why MLLMs struggle to generalize from in-context examples for this specific task? Is it a failure to grasp the "intent" behind the examples, or a difficulty in applying the complex, multi-faceted rules described in the prompt?
3.  The data was collected using "brand-new accounts with no browsing history" to avoid personalization bias. However, could this introduce a different kind of bias, such as over-representing content shown to new users? Do you anticipate differences in the distribution of covert ads seen by established versus new users, and how might this affect the generalizability of models trained on CHASM?
4.  The paper emphasizes the inclusion of challenging negative samples (genuine sharing posts). How were these "deceptive" negative samples specifically sourced or selected during the collection phase to ensure they were not just random non-promotional content? Was a specific keyword-based strategy employed to find posts that are highly product-related but likely non-commercial?

**Strengths Contributions:**

1.    The paper identifies and formally defines covert advertisement detection, an important but previously unexplored challenge in social media content moderation.
2.    CHASM provides carefully curated, privacy-protected data with systematic annotation guidelines and quality control measures for evaluating covert ad detection.
3.    The study tests 15 different MLLMs under multiple settings (zero-shot, in-context learning, fine-tuning) with detailed performance analysis.
4.    The work addresses a real-world problem with legal and ethical implications, providing concrete directions for improving social media content moderation.

---

> ### Author Rebuttal · Authors · 2025-07-29
>
> We sincerely appreciate your valuable time and insightful comments! We hope that our responses below will address your concerns.
>
> **About the dataset:**
>
> Regarding the definition of covert advertisement, we note that our formulation in Section 2.1 is grounded in prior marketing literature, as stated in the paper. Nonetheless, we acknowledge that further justification of its objectivity and universality would strengthen the framework. We will expand this discussion and consider consulting domain experts in the camera-ready version.
>
> For the error analysis, we appreciate the reviewer’s call for deeper reasoning. We will enhance Section 5.1 with additional insights—particularly to examine whether the model’s improved performance stems from better grounding in promotional intent or reliance on surface-level patterns. Lastly, we agree that the lack of user behavior and creator-level data is not only a future direction but also a notable limitation of the current study. We will revise the relevant sections to more directly acknowledge this limitation and discuss its impact on annotation ambiguity and label reliability.
>
> **About the experiment:**
>
> We also clarify that RedNote is an app originally built around shopping guides, which naturally leads to a higher concentration of product-related and advertorial content. Therefore, RedNote provides a more abundant and accessible source of relevant samples and represents a more pressing issue to address. Due to time constraints during the rebuttal phase and the varying data access policies across platforms (e.g., Weibo, Douyin), we were unable to incorporate additional data sources at this stage. In future work, we plan to expand our dataset to include a broader range of sources, such as multilingual and cross-cultural social media platforms, as well as a more diverse set of Chinese platforms.
>
> We agree that analyzing model performance across different types of covert advertisements (e.g., comment-based, image-based, or mixed signals) would provide valuable insight beyond modality ablation. We will incorporate this type-specific difficulty analysis in the final submission.
>
> **About the writing:**
>
> We acknowledge the inconsistencies and will correct them for improved clarity and academic tone in the camera-ready version.
>
> **Other Questions:**
>
> 1. We observed that the initial inter-annotator agreement, as measured by Fleiss' Kappa, was approximately 0.65. Notably, there were clear disagreements among annotators on certain challenging samples. To address this issue, we designed a dynamic annotation process combined with a majority voting strategy. We believe these measures help substantially reduce annotator disagreement and improve the overall consistency of the annotations. We will refine the wording in the camera-ready version.
>
> 2. Current MLLMs, when prompted via in-context learning, often rely on surface-level heuristics rather than truly reasoning about user intent or deceptive framing. Our task requires models to detect subtle, deliberately hidden commercial signals, which involves multi-modal reasoning, tone analysis, and recognizing indirect cues. These are hard to infer purely from examples, especially without gradient-based adaptation. Fine-tuning helps the model internalize these nuanced patterns, whereas in-context learning remains brittle in such settings.
>
> 3. For data collection, we acknowledge that using brand-new accounts may introduce a cold-start bias, but it offers the most neutral and controlled setup compared to personalized accounts shaped by prior activity.
>
> 4. Finally, for negative samples, we collected data via random crawling from new accounts, without any keyword filtering or manual search. This avoids introducing bias and ensures the negatives reflect naturally occurring, product-related but non-promotional content.
>
> Best regards,
>
> Authors

---

> > ### Comment · Reviewer_QrFi · 2025-08-04
> >
> > Thanks for your response, I will keep my score

---

### Official Review · Reviewer_MeEh · 2025-06-22

**Rating:** 4
**Confidence:** 5

**Summary:**

This paper presents a new dataset curated from the Chinese social media platform RedNote, designed to detect covert advertisements. The dataset includes 4,992 instances collected between September and October 2024, comprising both regular posts and those containing covert advertising content.

The authors follow a multi-step data collection and annotation process, including guideline development, annotator training, and multi-annotator labeling to ensure quality and consistency of ground truth labels. They benchmark a variety of state-of-the-art vision-language and language-only large models, spanning both public and proprietary LLMs, under zero-shot, few-shot, and fine-tuning settings.

Initial results show that zero-shot and few-shot models perform poorly on this task, highlighting the challenge and novelty of the dataset. However, with fine-tuning, Qwen, an open-source model, achieves the best performance, improving classification accuracy by approximately 30% compared to its zero-shot baseline.

Finally, the paper includes an error analysis to identify common failure cases among the evaluated models and provides discussion on potential future research directions informed by these insights.

**Dataset Code Accessibility:**

Yes

**Dataset Code Comments:**

The dataset is already published on HuggingFace, accompanied by clear documentation, metadata, and example usage code. The availability and clarity of these resources significantly support reproducibility and future adoption.

**Ethical Considerations:**

No, there are no or only very minor ethics concerns

**Final Justification:**

Considering the authors’ response, discussions with the authors, other reviewers, and the AC, the additional ablation studies have strengthened the completeness of the original work. However, the lack of evaluation on cross-lingual and cross-cultural generalizability remains a notable limitation, as such analysis is important for fully understanding the paper’s broader applicability. Therefore, I maintain my original assessment and recommend a borderline accept for this paper.

**Limitations Weaknesses:**

While the paper is strong overall, I see two areas where improvements or additional discussion would strengthen the work:

1. Generalization to English-language platforms. The dataset and experiments are focused entirely on Chinese social media. However, many state-of-the-art LLMs are predominantly trained on English corpora, making it unclear how the findings transfer to platforms such as Facebook, Instagram, or Threads. Including a small-scale cross-lingual experiment or qualitative examples showing how a model fine-tuned on the CHASM dataset performs on English content would add valuable insight. Quantitative results are ideal, but qualitative comparisons would still be helpful.

2. Cultural and regional generalizability. The paper does not discuss how cultural, ethical, or regional differences might influence the perception or effectiveness of covert advertising. For instance, what constitutes "covert" or misleading may vary significantly across different cultural or legal contexts. A brief discussion on how such factors might affect both dataset annotation and model performance would strengthen the broader applicability of the work. Analysis of potential divergence factors within the CHASM dataset (e.g., regional linguistic patterns) could also inform future directions.

**Strengths Contributions:**

This paper presents the first dataset focused on covert advertising, specifically targeting Chinese social media platforms. This is a timely and impactful contribution, given the growing concern around deceptive advertising practices in e-commerce. The paper addresses an important and underexplored problem with real-world implications for consumer protection and platform integrity.

The authors conduct a rigorous data collection and annotation process, followed by comprehensive benchmarking of multimodal large language models (MLLMs) under zero-shot, few-shot, and fine-tuning settings. The evaluation is thorough and well-structured, demonstrating clear technical depth and reproducibility.

The paper is well-written and clearly organized, making it easy to follow the motivation, methodology, and results. Figures and tables are informative and appropriately used to support the key claims. Notably, the dataset has already been made publicly available, which will greatly benefit future research in this area. Overall, this work makes a strong contribution to the NeurIPS community.

---

> ### Author Rebuttal · Authors · 2025-07-29
>
> We sincerely appreciate your valuable time and insightful comments! We hope that our responses below will address your concerns.
>
> **1. Generalization to English-language platforms.**
>
> We agree that cross-lingual generalization is important. During the rebuttal phase, we examined public Instagram datasets but found they lacked covert advertisements as defined in our paper. Most ads were overt ads with clear promotional tags (e.g., “#ad”, “#sponsored”) or direct sales content, and thus do not qualify as deliberately disguised.
>
> This may reflect cultural and regulatory differences—covert advertising appears to be more prevalent on Chinese platforms, while English platforms may adopt different native advertising patterns shaped by norms like FTC disclosure rules. Due to these gaps and limited time, we could not construct a suitable English test set. We plan to manually curate a small-scale English dataset aligned with our definition and report cross-lingual findings in the future work.
>
>
> **2. Cultural and regional generalizability.**
>
> Our definition focuses on the core characteristics of covert advertising, specifically clear commercial intent and deliberate disguise, which remain consistent across cultural and legal contexts. Such practices are widely regarded as deceptive and unethical, and are prohibited in jurisdictions including China, the European Union, and the United States. We reference both Chinese and U.S. regulations in Section 2.1 to support the generalizability of our criteria.
>
> Although these standards are broadly applicable, subtle differences may still exist in how covert advertising appears across different cultures and platforms. A more fine-grained cultural and regional analysis would require the collection of representative data from a wider range of cultural contexts. We plan to pursue such cross-cultural comparisons in future work.
>
> Best regards,
>
> Authors

---

> > ### Comment · Reviewer_MeEh · 2025-08-01
> >
> > Thank you for your thoughtful responses and for acknowledging the importance of cross-lingual and cross-cultural generalization. I understand the practical constraints around curating an English-language dataset and conducting a broader regional analysis within the rebuttal period. That said, I still believe these aspects are crucial for fully establishing the generalizability of the proposed framework, especially given the paper’s emphasis on covert advertising—a phenomenon that may manifest differently across linguistic and cultural environments.
> >
> > As you noted, even a small-scale English dataset or preliminary cross-cultural analysis could have offered valuable insights. I appreciate your plan to address these in future work, but the absence of any supporting empirical evidence in this direction slightly limits the current paper’s broader applicability.
> >
> > For these reasons, I will maintain my original score. I hope the authors continue this important line of research and look forward to future work that expands its cross-lingual and cross-cultural scope.

---

### Official Review · Reviewer_P8Gn · 2025-07-01

**Rating:** 4
**Confidence:** 4

**Summary:**

This paper introduces CHASM, a novel dataset focused on detecting covert advertisements on Chinese social media, specifically RedNote (i.e., Xiaohongshu). The dataset includes 4,992 multimodal (text-image) posts labeled for covert advertising, with experiments evaluating the performance of MLLMs under various learning settings (zero-shot, in-context, fine-tuning). The authors argue for the importance of this underexplored yet socially impactful task, offering a benchmark that poses subtle semantic and visual challenges to existing models.

**Dataset Code Accessibility:**

Partly

**Ethical Considerations:**

No, there are no or only very minor ethics concerns

**Final Justification:**

My concerns were sufficiently addressed in the rebuttal.

**Limitations Weaknesses:**

1. The definition of “covert advertisement” is not formalized sufficiently. While the paper describes some qualitative features (e.g., persuasive tone, pseudo-authentic reviews), it lacks rigorous operational criteria to distinguish covert ads from, say, enthusiastic user reviews. Without a clearer taxonomy of deception (e.g., intent-based vs. outcome-based definitions), the validity of the labels remains somewhat subjective.
2. Although the paper claims “strict quality control protocols,” it does not provide annotation guidelines, inter-annotator agreement metrics, or the annotator backgrounds. Given the subtlety of the task, these details are critical. For instance, how were borderline cases resolved? Were annotators trained or domain experts?
3. The evaluation of MLLMs is limited to performance metrics without any statistical tests or comparisons with simpler baselines (e.g., logistic regression over TF-IDF features, or ResNet + BERT). Moreover, the in-context learning setting is not clearly described: What prompts were used? How were examples selected? These choices can significantly affect performance and should be reproducible.
4. While the authors demonstrate that fine-tuning improves performance, they do not offer any analytical insights into what makes a sample “hard” or “easy” for the model. An error taxonomy or attention heatmaps could help understand whether models fail due to linguistic subtlety, visual misdirection, or multimodal incongruence.
5. The dataset is available on HuggingFace, but there are issues with missing Responsible AI documentation (e.g., annotator demographics, bias mitigation, downstream use). Also, some files in the dataset viewer are inaccessible, which needs to be fixed to ensure usability.

**Strengths Contributions:**

1. The paper tackles an important and timely problem. Covert advertisements are indeed harder to detect than explicit ones and are pervasive in influencer marketing, especially in Chinese social media.
2.  The inclusion of both image and text makes the task more realistic and challenging, aligning well with practical detection scenarios.
3. The authors benchmark a wide range of MLLMs and show that current models perform poorly, highlighting a gap between research capability and real-world demand.

---

> ### Author Rebuttal · Authors · 2025-07-29
>
> We sincerely appreciate your valuable time and insightful comments! We hope that our responses below will address your concerns.
>
>
> **1. The definition of “covert advertisement” is not formalized sufficiently.**
>
> We acknowledge that covert advertising is inherently subjective, and we try our best to have a precise and operational definition. As detailed in **Section 2.1 and Appendix A**, we adopt an evidence-driven guideline, requiring both clear commercial intent and deliberate disguise.
> Specifically, we elaborate this into an evidence-driven guideline with 13 concrete criteria, such as promotional tone, hidden purchase links, brand emphasis in images, and alignment between text and product visuals. We also provide multiple positive and negative examples to guide annotators in distinguishing covert ads from genuine user posts. These measures enable consistent labeling in practice, as demonstrated by the 94% accuracy on gold-standard questions.
>
> **2. Annotation detail.**
>
> We provide detailed annotation guidelines in Appendix A, including a formal task definition, 13 evidence-based criteria, and examples to guide consistent labeling. In Section 3.2, we describe our three-stage annotation workflow, which includes annotator training, qualification tests using gold-standard questions, and a dynamic voting mechanism for difficult cases. Rather than inter-annotator agreement, we report 94% accuracy on hidden gold questions, which is more appropriate under our non-static annotation assignment. Annotators were trained native Chinese speakers with RedNote experience, ensuring domain familiarity.
>
> **3. Statistical test, simpler baseline, and prompt.**
>
> We include statistical significance tests in our evaluation—Table 3 reports p-values (< 0.01), confirming that fine-tuning significantly outperforms zero-shot baselines. In response to the review, we also added new experiments with simpler models, including TF-IDF + LR and BERT + ResNet:
>
> | Model         | Accuracy | Precision | Recall | F1 Score |
> | ------------- | -------- | --------- | ------ | -------- |
> | TF-IDF + LR   | 64.7%    | 65.5%     | 64.8%  | 64.4%    |
> | BERT + ResNet | 65.3%    | 72.2%     | 65.3%  | 61.5%    |
>
> All baselines were trained using the same setting as the we fine-tuning MLLMs. Notably, these models achieve significantly lower performance than fine-tuned MLLMs (e.g., Qwen2.5-7B: F1 = 75.6%), further illustrating the complexity and subtlety of this task.
>
> For in-context learning, we already provided full prompt templates in Appendix C and D, ensuring reproducibility.
>
> **4. Error taxonomy.**
>
> We provide a detailed **error taxonomy** in **Section 5.1 and Table 4**, which goes beyond a binary “hard vs. easy” categorization. Our taxonomy includes **four interpretable error types**: *Insufficient Evidence*, *Missing Clue* (e.g., overlooked image or comment signals), *Style Misjudgment* (e.g., confusion over persuasive tone), and *Pattern Confusion* (e.g., structural overlap with genuine sharing). This categorization allows for a more **granular understanding of failure modes** across models and modalities, and offers more actionable insight than a simple difficulty split.
>
> After fine-tuning, we observe a clear shift: “Insufficient Evidence” errors decrease significantly, indicating that the model becomes better at aligning with human-like, evidence-based standards for detecting covert intent.
>
> **5. More details about HuggingFace documentation.**
>
> We have carefully reviewed the HuggingFace repository and confirmed that it is fully accessible, as also verified by other reviewer (MeEh). If there are specific issues, we welcome further clarification. Additionally, we will add more documentation—such as annotator demographics and usage notes—to further improve usability and transparency before the camera-ready version.
>
>
>
>
> Best regards,
>
> Authors

---

> > ### Comment · Reviewer_P8Gn · 2025-08-06
> >
> > Thank you for the response. It clarified my concerns, and I have accordingly increased my score.

---

> ### Author Response · Authors · 2025-08-05
>
> Dear Reviewer P8Gn,
>
> Thank you again for your insightful review of our paper! We are writing to follow up on our rebuttal, we provide a brief summary of the clarifications in our rebuttal addressing your comments, and we hope this will help foster a clearer understanding of our work:
>
> - **Definition of “covert advertisement” and annotation detail:** As detailed in Section 2.1 and Appendix A, we define covert advertisements as content with both commercial intent and deliberate disguise, operationalized via 13 clear criteria and annotated examples. Our systematic guidelines and rigorous annotation protocol were recognized by reviewers QrFi and b1kH.
>
> - **Statistical Significance and Simpler Baselines:**  Table 3 reports statistically significant improvements (p < 0.01) of fine-tuned models over zero-shot baselines. We also tested simpler models (e.g., TF-IDF + LR, BERT + ResNet), which performed markedly worse (e.g., F1 = 64.4% vs. 75.6% for Qwen2.5-7B), highlighting the task’s complexity and the need for fine-tuning.
>
> - **Error taxonomy:** Section 5.1 and Table 4 present a four-category error taxonomy, offering interpretability beyond binary “hard vs. easy” splits. Fine-tuning notably reduces “Insufficient Evidence” errors, indicating improved alignment with human evidence-based judgment standards.
>
> - **Dataset Availability:** We have carefully reviewed the HuggingFace repository and confirmed that it is fully accessible, as also verified by reviewer MeEh.
>
>
>
> We would also like to highlight that several reviewers have positively recognized the key contributions and strengths of our work:
>
> - **Novelty of the task:** All reviewers recognized our paper for introducing a novel and timely task: covert advertisement detection, addressing a critical yet understudied issue.
>
> - **High-quality dataset and comprehensive experiments:**  Reviewers MeEh, QrFi and b1kH appreciated the carefully curated, privacy-protected dataset with systematic annotation guidelines. They also noted that the experiments are comprehensive, effectively showing that even state-of-the-art models currently struggle with the challenges posed by our proposed task.
>
> - **Clear writing and organization:** Reviewers MeEh and b1kH highlighted that the paper is well-written, clearly organized, easy to follow, and highly engaging.
>
> We would therefore be very grateful if you would consider re-evaluating our submission and raising your score better to reflect its novelty and importance to the field. Please feel free to reach out if you have any further questions or would like to discuss any aspect in more detail.
>
>
> Best
>
> Authors

---

### Note · Authors · 2025-08-12

We sincerely thank the reviewers for their constructive feedback and recognition of our work. Across the reviews, the novelty, timeliness, and societal impact of **CHASM**, the first multimodal dataset for covert advertisement detection, were consistently recognized.

**Reviewers noted in particular**:

1. **Novel and important problem**: Covert advertisements are pervasive yet underexplored, posing practical challenges for content moderation.
2. **High-quality dataset**: Systematic annotation guidelines, privacy protection, and rigorous quality control.
3. **Comprehensive experiments**: Benchmarks with 15 MLLMs under zero-shot, in-context, and fine-tuning settings, accompanied by an interpretable error taxonomy.
4. **Clear presentation**: Well-written, well-organized, and engaging.

We appreciate the reviewers’ concerns and have provided further clarification on their misunderstandings:

In our rebuttal, **we clarified**:

- **Definition & Annotation Protocol**: We further clarify that our work follows a rigorous annotation protocol and a clear task definition. The paper already includes a formal definition, 13 criteria with illustrative examples, and detailed workflow descriptions; In the rebuttal, we have elaborated on these aspects to avoid misunderstandings.

- **Error Taxonomy**: In the rebuttal, we clarified the four-category error taxonomy presented in the paper, offering interpretability beyond a binary “hard vs. easy” split. Fine-tuning notably reduces “Insufficient Evidence” errors, indicating improved alignment with human evidence-based judgment standards.

- **Reason for Data Source Selection**: We also clarify that RedNote is an app originally built around shopping guides, naturally resulting in a higher concentration of product-related and advertorial content. Therefore, RedNote offers a more abundant and accessible source of relevant samples.



**Planned Camera-Ready Improvements**:

- More explicit discussion of the task’s subjectivity.

- Added comparison results with simpler baselines: TF-IDF + LR and BERT + ResNet.

- Improved writing clarity to avoid potential misunderstandings (e.g., clearer explanation of Table 1).

- Responsible AI documentation on the dataset release.

We believe these improvements will further strengthen the paper’s contribution and reproducibility, and we are committed to continuing this research toward broader language and platform coverage, as well as larger-scale datasets.

---

### Decision · Program_Chairs · 2025-09-18

**Decision:**

Accept (poster)

**Comment:**

This paper introduces a novel dataset focused on detecting covert advertisements on Chinese social media, specifically RedNote (i.e., Xiaohongshu). The reviewers found the paper to be novel, well-written, and addresses an important and underexplored problem. Most of the reviewers' concerns were satisfactorily addressed in the rebuttal. However, the authors are encouraged to incorporate the reviewers' suggestions, as well as the Planned Camera-Ready Improvements Author Final Remarks, in the final version.

===== FINAL UPDATE FROM DB Track PCs ====

The final decision for this paper has been taken by the program chairs after consultation with the SACs. All Senior Area Chairs have ranked papers according to the feedback from the AC during the review process. We decided to leave the original meta-review to reflect the opinion of the AC in light of the initial discussions with reviewers and SAC.